# GyrI-like proteins catalyze cyclopropanoid hydrolysis to confer cellular protection

Hua Yuan [1], Jinru Zhang[1], Yujuan Cai[1], Sheng Wu[1], Kui Yang[1], H.C. Stephen Chan[2], Wei Huang[1], Wen-Bing Jin[1], Yan Li[1], Yue Yin[1], Yasuhiro Igarashi[3], Shuguang Yuan [4], Jiahai Zhou[1] & Gong-Li Tang [1]

GyrI-like proteins are widely distributed in prokaryotes and eukaryotes, and recognized as small-molecule binding proteins. Here, we identify a subfamily of these proteins as cyclopropanoid cyclopropyl hydrolases (CCHs) that can catalyze the hydrolysis of the potent DNA-alkylating agents yatakemycin (YTM) and CC-1065. Co-crystallography and molecular dynamics simulation analyses reveal that these CCHs share a conserved aromatic cage for the hydrolytic activity. Subsequent cytotoxic assays confirm that CCHs are able to protect cells against YTM. Therefore, our findings suggest that the evolutionarily conserved GyrI-like proteins confer cellular protection against diverse xenobiotics via not only binding, but also catalysis.

[1] State Key Laboratory of Bio-organic and Natural Products Chemistry, Shanghai Institute of Organic Chemistry, Chinese Academy of Sciences, 345 Lingling Road, Shanghai 200032, China. [2] Faculty of Life Sciences, University of Bradford, Bradford, West Yorkshire BD7 1DP, UK. [3] Biotechnology Research Center, Toyama Prefectural University, 5180 Kurokawa, Imizu, Toyama 939-0398, Japan. [4] Laboratory of Physical Chemistry of Polymers and Membranes, Ecole Polytechnique Fédérale de Lausanne (EPFL), CH B3 495 (Bâtiment CH) Station 6, CH-1015 Lausanne, Switzerland. Hua Yuan and Jinru Zhang contributed equally to this work. Correspondence and requests for materials should be addressed to S.Y. (email: shuguang.yuan@gmail.com) or to J.Z. (email: jiahai@sioc.ac.cn) or to G.-L.T. (email: gltang@sioc.ac.cn)

Cyclopropanoid natural products possess a common-strained, three-membered cyclopropane ring, which frequently show excellent biological activities, and may serve as potential drug leads[1, 2]. Yatakemycin (YTM, **1**), CC-1065 (**2**), and duocarmycins (**3** and **4**) belong to a family of cyclopropapyrroloindole compounds that exhibit remarkably potent DNA-alkylating activities[3, 4] (Fig. 1a); these have been harnessed to develop antibody-drug conjugates[5]. For example, the duocarmycin-derived ADC MDX-1203 has entered Phase I clinical trials for treating non-Hodgkin lymphoma and renal cell carcinoma[5]. Microorganisms that synthesize these cytotoxic cyclopropanoids must evolve efficient strategies to avoid suicide. Exploration of resistance genes from their producers is not only beneficial for drug discovery, but also for assessing the lifespan of drug utility[6–8]. Our previous biosynthetic studies on **1** revealed that a DNA glycosylase YtkR2 repairs the N3-YTM-alkylated adenine residues and confers self-resistance against **1** in the last resort[9, 10]. Here, we show an additional level of resistance regarding YTM cyclopropyl hydrolysis catalyzed by a subfamily of GyrI-like proteins exemplified by YtkR7, a gene product coded for within the same biosynthetic gene cluster of **1**. We define this subfamily of GyrI-like proteins as cyclopropanoid cyclopropyl hydrolases (CCHs), and demonstrate that they share a conserved aromatic cage for the hydrolysis of **1** and **2**, and thereby confer cellular protection. The results presented here suggest that the evolutionarily conserved GyrI-like proteins are involved in cellular detoxification against diverse xenobiotics via not only binding, but also catalysis.

## Results

**Identification of a subfamily of GyrI-like proteins as CCHs.** Bioinformatic analysis showed that YtkR7 bears high homology to the GyrI-like small-molecule binding domain (SMBD), members of which share sequence similarity with SbmC (also designated as GyrI)[11], a protein that can protect *Escherichia coli* from DNA replication inhibitors (e.g., microcin B17[12]) and DNA-damaging agents (e.g., mitomycin C[13]). The GyrI-like proteins possess a duplicate βαββ configuration and appear to have been adapted for small-molecule binding[11]. However, the limited knowledge about the binding properties of these proteins is mainly gained from the SMBD of the MerR-like transcription activator BmrR, which can accept structurally diverse inducers to control the expression of the multidrug transporter Bmr[11, 14–19].

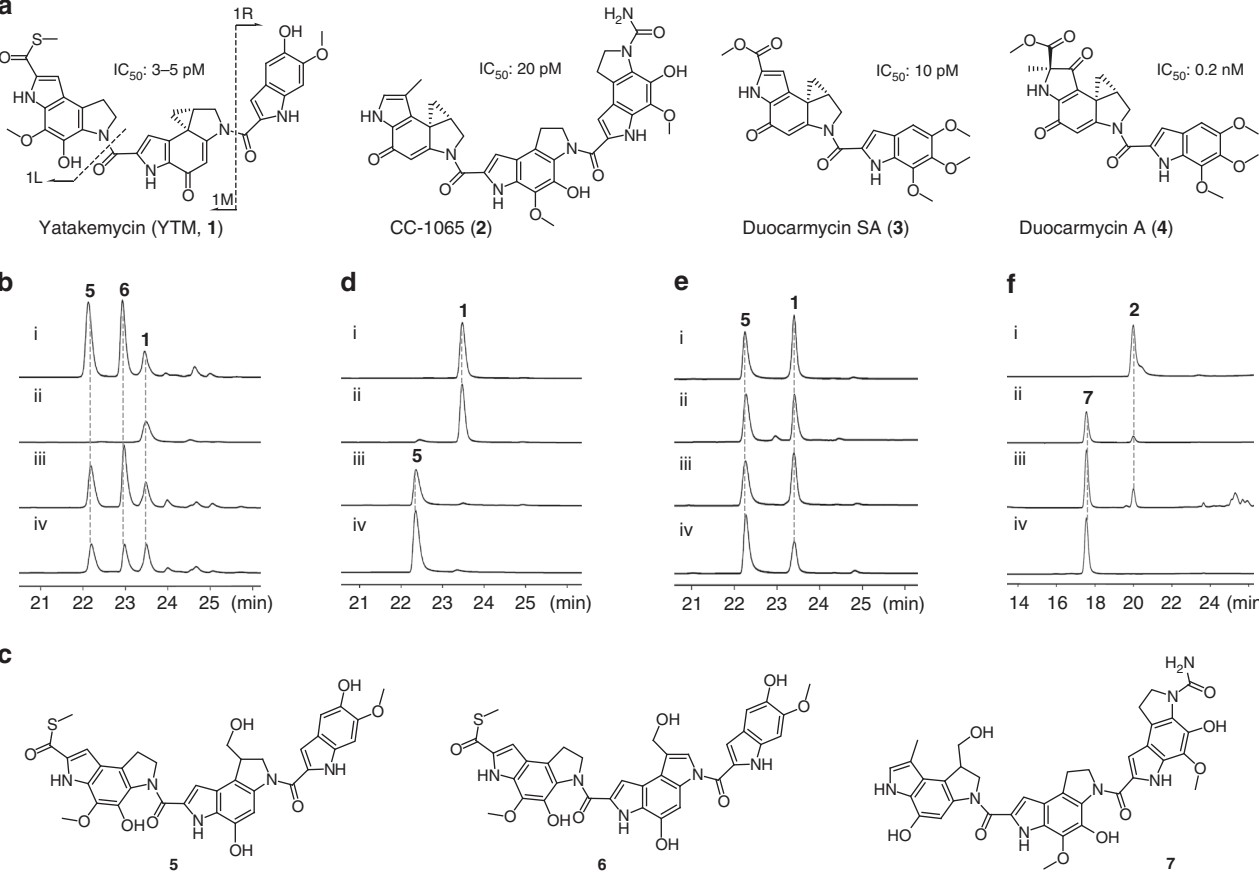

**Fig. 1** Structures of cyclopropapyrroloindole compounds and characterization of cyclopropanoid cyclopropyl hydrolases (CCHs). **a** Structures of yatakemycin (YTM), CC-1065, and duocarmycins. 1 L, 1 M, and 1 R indicate the three subunits of YTM, respectively. The IC$_{50}$ reveals the cytotoxicity against L1210 cell line. **b** Genetic investigation of *ytkR7* by HPLC analysis of the fermentation products (UV at 383 nm). (i) wild-type *Streptomyces* sp. TP-A0356; (ii) the Δ*ytkR7* mutant *Streptomyces* sp. TG1310; (iii) the Δ*ytkR7* mutant complemented with the *ytkR7* gene in trans; and (iv) the Δ*ytkR7* mutant complemented with the *c10R6* gene from *S. zelensis* NRRL 11183. **c** Structures of compounds **5**, **6**, and **7**. **d** Biochemical characterization of YtkR7 with YTM as substrate. (i) YTM dissolved in the reaction buffer; (ii) boiled YtkR7; (iii) YtkR7; and (iv) standard of **5**. **e** Biochemical assays of the other selected CCHs using YTM as substrate. (i) C10R6; (ii) lin2189 from *Listeria innocua* Clip11262; (iii) ETI84332.1 from *Streptococcus anginosus* DORA_7 (from the human microbiota); and (iv) MA1133 from *Methanosarcina activorans* C2A. **f** Characterization of substrate specificity of the CCH protein C10R6 by HPLC analysis (UV at 374 nm). (i) CC-1065 dissolved in the reaction buffer; (ii) C10R6 with CC-1065 as substrate; (iii) the fermentation products of *S. zelensis* NRRL 11183; and (iv) standard of **7**.

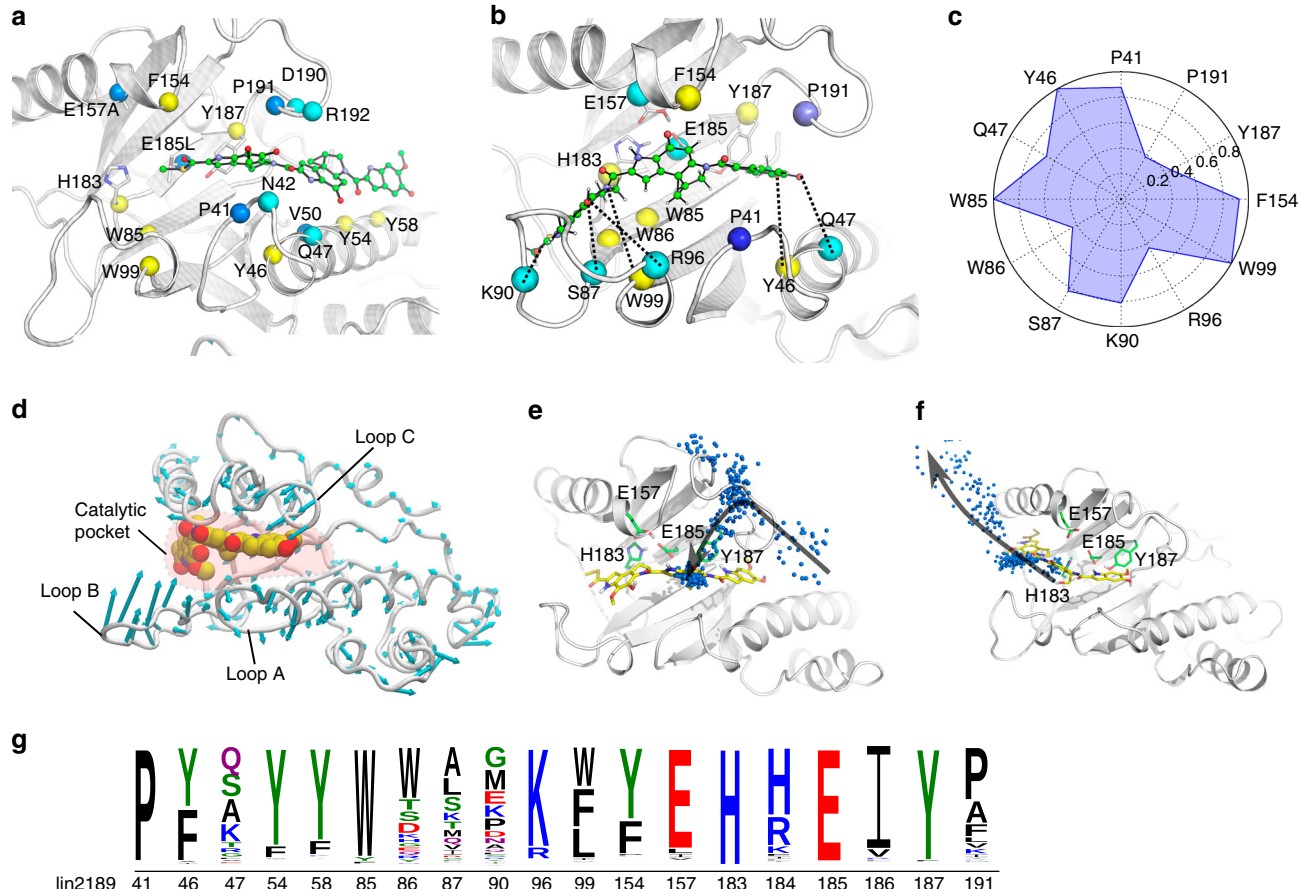

**Fig. 2** Structural and molecular dynamics (MD) simulation analyses of lin2189-YTM. **a** The molecular details of the lin2189 E157A/E185L-YTM crystal structure. Green ball-and-stick cartoon, the substrate (YTM); cyan spheres, polar residues; yellow spheres, aromatic residues; and blue spheres, hydrophobic residues. **b** Interactions between native lin2189 and YTM from MD simulations. Black dash, H-bond interactions. **c** The normalized interaction frequency of native lin2189-YTM calculated from unbiased MD simulations. **d** The intrinsic mobility of lin2189 calculated from the principal component analysis (PCA). The cyan vector length correlates with the domain-motion scale. Yellow sphere, the substrate molecule (YTM) from docking; red region, the catalytic pocket. PCA analysis showed that loops A, B and C next to the catalytic pocket are very flexible. **e** The substrate entrance pathway sampled by metaMD simulations. Yellow stick, final pose of the substrate YTM. Green stick, the catalytic trait in the binding pocket. Blue spheres, the mass center of the substrate molecule along the simulation trajectory. **f** The product leaving pathway sampled by metaMD simulations. **g** The catalytic residues and aromatic cage of CCHs are highly conserved. YtkR7, and the 1695 similar proteins were used to create the sequence logo. The residue numbers correspond to those of lin2189

The vast number of GyrI-like sequences in databases (12,304 sequences as of September 2016 (Supplementary Fig. 1)) from both prokaryotes and eukaryotes suggest that these proteins perform some evolutionarily conserved biological functions. To investigate the function of YtkR7, we inactivated the *ytkR7* gene and obtained the *ΔytkR7* mutant TG1310 (Supplementary Fig. 2). Comparison of the metabolite profiles of the wild type with the mutant showed that the inactivation of *ytkR7* did not affect the production of **1**, but two compounds (**5** and **6**) that were present in the wild type disappeared in TG1310 (Fig. 1b, ii). Additionally, complementation of native *ytkR7* in trans into TG1310 could restore the generation of **5** and **6** (Fig. 1b, iii). We then isolated and characterized both as YTM cyclopropyl hydrolyzed derivatives (Fig. 1c and Supplementary Figs 3–6). Compound **6** was later found to be a spontaneously oxidized product from **5**. The unexpected structure of **5** led us to speculate that YtkR7 is an enzyme capable of catalyzing YTM cyclopropyl hydrolysis. To test this hypothesis, we expressed and purified YtkR7 from *E. coli* (Supplementary Fig. 7). As expected, incubation of YtkR7 with **1** afforded **5**, which was not observed for the control assay (Fig. 1d). Notably, both a high concentration of DMSO and acidic buffers promote the enzymatic hydrolysis activity (Supplementary Fig. 8).

Collectively, our genetic and biochemical results demonstrate that YtkR7 functions as a CCH.

Using YtkR7 as a query, 1695 similar proteins found exclusively in prokaryotes[20] were retrieved to generate a sequence similarity network (Supplementary Fig. 9). Seven of these GyrI-like proteins were chosen from a variety of bacteria and archaea (from diverse environments, including soil and the human microbiota: SHJG_8481, Chte2144, ETI84332.1, C10R6, lin2189, SSDG_03674, and MA1133) and tested for their enzymatic activity, and all were found to convert **1** to **5** (Fig. 1e; Supplementary Figs 10 and 11), suggesting that a great number of prokaryotic GyrI-like proteins can be classified as CCHs.

When we tested the enzymatic activity of C10R6 in the mutant strain TG1310, we found that it can restore the production of **5** and **6** in vivo (Fig. 1b, iv). Given the fact that C10R6 is encoded by the biosynthetic gene cluster of **2**[21], this protein may function to catalyze CC-1065 cyclopropyl hydrolysis. Indeed, incubation of C10R6 with **2** showed that C10R6 could convert **2** to a new derivative **7** (Fig. 1f, ii). This derivative was also present in the strain that produces **2** (Fig. 1f, iii); isolation and structural elucidation revealed **7** to be a CC-1065 cyclopropyl hydrolyzed product (Fig. 1c; Supplementary Figs 12 and 13). Next, **2** was used

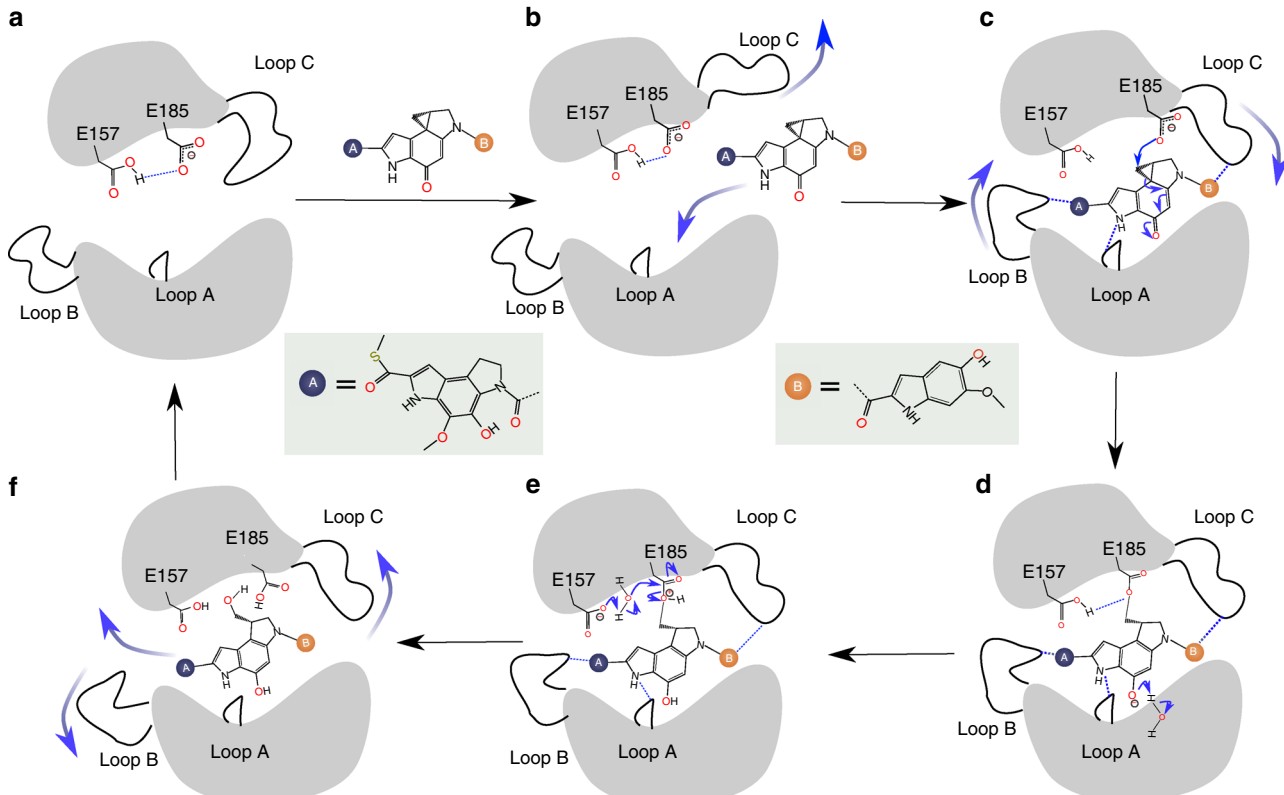

**Fig. 3** The catalytic mechanism of CCHs. **a** In an apo CCH, the three loops (loop A, B, and C) next to the catalytic pocket are very flexible and the size of the pocket can be fluctuated to a large extent. E157 is protonated, whereas E185 is deprotonated. **b** The entry of substrate into the CCH leads to the outward shift of loop C, resulting in a large space in the binding pocket. This facilitates the substrate molecule entering the catalytic region. **c** After substrate situating at the catalytic site and in an activated binding mode. E185 acts as a nucleophile, attacking the methylene group in the cyclopropyl ring. This leads to transformation of the quinone moiety into a deprotonated phenol. **d** The deprotonated phenol moiety of the transitional substrate acquires a proton from a water molecule. **e** With the assistance of E157, the transitional state is hydrolyzed by a water molecule. **f** Finally, with the outward fluctuations of loop B and loop C, the product leaves catalytic pocket. The E185 residue returns to its deprotonated state

to probe the substrate specificity of YtkR7 and the other selected CCHs, and in vitro enzymatic assays showed that SSDG_03674, lin2189, Chte2144, and SHJG_8481 could hydrolyze **2** (Supplementary Fig. 14), while YtkR7, ETI84332.1, and MA1133 could not. Our in vitro and in vivo studies showed that all of the tested CCHs could hydrolyze **1**, but were variously promiscuous with other structurally related substrates.

**Structural and MD simulation analyses of CCHs**. To probe the catalytic mechanism of CCHs, we carried out co-crystallography and molecular dynamics (MD) simulation experiments on lin2189, a CCH protein from *Listeria innocua* that can hydrolyze both **1** and **2**. The apo structure of lin2189, which was solved previously (PDB ID: 3B49), is comprised of three α-helices and six β-strands (Supplementary Fig. 15). Co-crystallization of native lin2189 with **1** or **5** turned out to be extremely difficult, owing, respectively, to rapid hydrolysis and low affinity. Previous structural and biochemical studies of BmrR revealed that the glutamate residue Glu-253 has a noticeable effect on its ligand binding and transcription activation[17]. This acidic residue is highly conserved among the GyrI-like proteins[11], we thus constructed a lin2189 variant (E185L). Interestingly, this variant lost the enzymatic activity and allowed us to obtain a complex structure of lin2189 E185L and **1** in which the electron density for the substrate is partially visible. Detailed investigation of the electron density map implicated that several residues in the

solvent-accessible channel (e.g., E157) potentially interfered with substrate binding. Therefore, we designed a series of variants based on E185L. As expected, we successfully solved the complex structure of a catalytically deficient form lin2189 E157A E185L and **1** with complete density at a resolution of 1.2 Å. The structure is composed of two monomers, each of which binds one molecule of **1** at the crevice. The 1L- and 1M-subunits of **1** are stretched into an aromatic cage consisting of Y46, W85, W99, F154, and Y187, while the 1R-subunit is exposed to the solvent and stabilized by Y54 and Y58 through π–π stacking interactions (Fig. 2a; Supplementary Figs 16a and 17). Surprisingly, we did not observe any catalytic residues close to the cyclopropane moiety, which is contrary to canonical ring opening mechanisms that require general acid/base[1, 22].

Subsequently, we used a protein–ligand docking approach to computationally construct the lin2189-**1** complex structure and validated our model with MD simulations. In the lin2189-**1** model, **1** was in the same aromatic cage observed in the co-crystal structure. However, the reactive methylene group in the cyclopropyl ring was next to the acidic residue E185 (4.5 Å) that is very close to another acidic residue E157 (2.8 Å) (Fig. 2b, c; Supplementary Fig. 16b), providing a likely electrostatic mechanism to explain both our difficulty in crystallizing the native complex and our ability to crystallize the double-mutant form. Consistent with the binding model for lin2189-**1**, we found that substrate **2** fit well in the same aromatic cage as **1** in the lin2189-**2** model (Supplementary Fig. 18). We next conducted well-

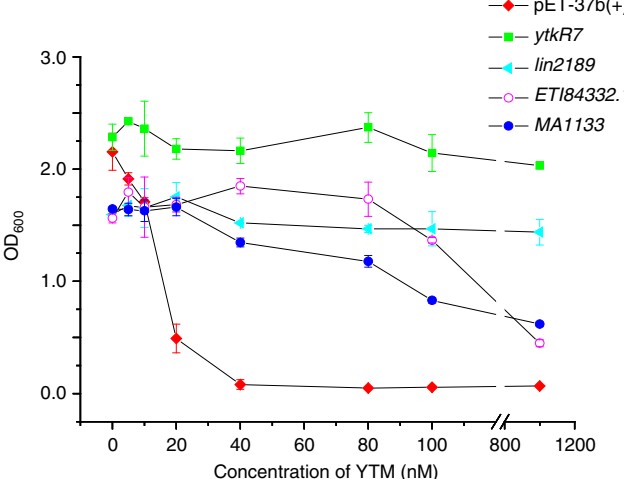

**Fig. 4** CCHs can protect *E. coli* against the potent DNA-alkylating agent YTM. Four selected CCH proteins include YtkR7, lin2189, ETI84332.1, and MA1133, which cover from both bacteria and archaea (from diverse environments, including soil and the human microbiota)

tempered metadynamics simulation (metaMD) to identify the substrate entrance pathway for **1**. Principal component analysis showed that both loop B and loop C undergo noticeable fluctuations (Fig. 2d; Supplementary Fig. 19 and Supplementary Movie 1). We therefore chose the mass center distances between loop B and loop C and between the substrate and residue E185, as the collective variables (CVs) for these simulations. During the metaMD simulation of the lin2189-**1** complex, the substrate first approached the vicinity of Y187 next to loop C. Next, loop C shifted outward, yielding a large space in the pocket suitable for substrate binding (Fig. 2e and Supplementary Movie 2). Such large movements are probably due to the induced fit effect which has been observed in various biological processes[23, 24]. Notably, the position of the complex crystal structure has been observed in the snapshots of our metaMD simulation (Fig. 2e and Supplementary Movie 2), which further confirms our identification of this substrate entrance. An additional metaMD simulation aiming to explore the product exit (with the same CVs) revealed that both loop B and loop C move outward markedly and suggested that the product leaves the binding pocket from the vicinity of loop B (Fig. 2f and Supplementary Movie 3).

The importance of the catalytic pocket to hydrolysis activity is further strengthened by the fact that the catalytic residues are very highly conserved among CCHs (Fig. 2g), and mutagenesis experiments confirmed that the acidic residues E157 and E185 are essential for the catalysis (Supplementary Fig. 20). The conserved aromatic residues W85 and Y187 in the binding pocket also have critical effects on the hydrolytic activity. Consistent with these findings, distantly related GyrI-like proteins lacking the pair of acidic residues and aromatic cage (e.g., SbmC[12], BmrR[14], and Rob[25]) were not able to hydrolyze **1** to **5** (Supplementary Fig. 21). Based on these data, we propose the following catalytic mechanism for CCHs (Fig. 3). In apo CCHs, E157 is protonated and E185 is deprotonated, due to the fact that both of them are buried in the deep pocket and engage with each other via an H-bonding contact. This has also been confirmed by pKa calculations on the MD trajectory (Supplementary Fig. 22). As the substrate enters the CCH, it leads to an outward shift of loop C, which facilitates further entry into the catalytic pocket. After the substrate is situated in the pocket in an activated binding mode, E185 attacks the methylene group in the cyclopropyl ring. This leads to transformation of the quinone

moiety into a deprotonated phenol, which acquires a proton from water. With the assistance of E157, the transitional substrate is then hydrolyzed by a water molecule. The hydroxyl group from water is transferred to E185, and a proton is gained by E157. Finally, outward fluctuations of loop B and loop C facilitate product exit from the catalytic pocket.

**CCHs protect cells against the DNA-alkylating yatakemycin.** Considering that the cytotoxic activity of **1** is attributed to its cyclopropane warhead[3], we suspected that **5** would not exhibit any biological activity. Indeed, in vitro activity assays demonstrated that **5** is not toxic to Jurkat cells or *E. coli* (Supplementary Fig. 23a, b). Furthermore, bacterial growth inhibition assays confirmed that *E. coli* cells expressing *ytkR7*, *lin2189*, *ETI84332.1*, or *MA1133* are much more tolerant to **1** than the control strain (Fig. 4 and Supplementary Fig. 23c). Domain analysis of the 12,304 GyrI-like sequences revealed that 4706 proteins consist of only a GyrI-like domain (e.g., SbmC[12] and the tested CCHs) according to the InterPro database[20] (Supplementary Fig. 24). The majority of the GyrI-like sequences are either AraC-like (5498; e.g., Rob[25]) or MerR-like (1824; e.g., BmrR[14]) transcriptional regulators. Although SbmC and Rob have been reported to be involved in cellular protection against toxic chemicals, their ability to bind small molecules is not as yet confirmed. Our further binding assays on 17 antibiotics showed that SbmC and Rob, as well as YtkR7, could bind 3–6 antibiotics (Supplementary Fig. 25). The finding that these distantly related GyrI-like proteins each have the ability to bind xenobiotics suggests that using the GyrI-like proteins for cellular detoxification appears to be an ancient strategy.

## Discussion

The vast distribution of GyrI-like proteins from both prokaryotes and eukaryotes suggests that they are much more like evolutionarily conserved proteins rather than those acquired by horizontal gene transfer, and thus they should play beneficial roles to their hosts. In this study, we have identified a subfamily of prokaryotic GyrI-like proteins as CCHs that can hydrolyze the cyclopropyl moieties of YTM and CC-1065, and thereby confer cellular resistance against these cytotoxic cyclopropanoids. Thus, these GyrI-like proteins directly function as antibiotic resistance proteins. Moreover, in vitro binding assay confirms that GyrI-like proteins each still retain the ability to bind several structurally diverse xenobiotics (Supplementary Fig. 25). Our results also provide strong support for the very recent proposal that GyrI-like proteins have potential multidrug resistance functions[26]. Structural studies of BmrR for drug binding and multispecificity reveal that aromatic side chains lining the drug pocket form a rigid, aromatic platform for diverse drugs to dock[18]. Subsequent structural comparison of lin2189, BmrR, SbmC, and Rob revealed that they each possess a different solvent-accessible cage formed by a set of aromatic and hydrophobic residues, which might in turn determine the ligand specificity of GyrI-like proteins (Supplementary Figs 26 and 27). Therefore, it can be concluded that GyrI-like proteins may be regarded as an intracellular scavenger for xenobiotics, and that organisms harness GyrI-like proteins for cellular protection as an ancient strategy.

Currently, antibiotic resistance is one of the major global concerns. Studies on antibiotic resistance genes located within their biosynthetic gene clusters will facilitate our understanding about the antibiotic resistome in the environment. Therefore, in addition to shedding new light on the vast capacity of organisms to deal with various toxic xenobiotics, our findings are very promising for future rational drug discovery.

## Methods

**Materials.** Biochemicals and media were purchased from Sinopharm Chemical Reagent Co., Ltd (China), Oxoid Ltd (UK), Sigma-Aldrich Corporation (USA), and Sangon Biotech Shanghai Co., Ltd (China), unless otherwise stated. Enzymes used for genetic manipulation were purchased from Thermo Fisher Scientific Co., Ltd (USA), Takara Biotechnology (Dalian) Co., Ltd (China) and New England Biolabs (USA). Enzymes used for PCR amplifications and mutagenesis were Taq DNA polymerase (Vazyme Biotech Co., Ltd) and Phanta Max Super-Fidelity DNA Polymerase (Vazyme Biotech Co., Ltd). The primers and synthetic genes (*Chte2144*, *ETI84332.1*, *lin2189*, *SSDG_03674*, *MA1133*, and *bmrR* (BSU24020)) were synthesized by GENEWIZ (China). Chemical reagents were purchased from standard commercial sources. All the primers, strains, and plasmids used are in Supplementary Tables 1 and 2, respectively.

**Sequence analyses of GyrI-like proteins.** The GyrI-like SMBD-containing proteins (totally 12,304 sequences up to September 2016) were obtained using YtkR7 as query from the InterPro website (http://www.ebi.ac.uk/interpro/search/sequence-search)[20]. PSI-BLAST using the human TEX264 (NCBI Reference Sequence: NP_001123356.1) as query was performed on the NCBI website (https://blast.ncbi.nlm.nih.gov/Blast.cgi?PROGRAM = blastp&PAGE_TYPE = BlastSearch&LINK_LOC = blasthome) to obtain homologous sequences. All the sequences were first clustered using BLASTClust on the website (https://toolkit.tuebingen.mpg.de/blastclust)[27]. The parameters included: (1) sequence length to be covered, 75% and (2) percent identity threshold, 40%. As a result, 907 clusters were obtained, and the representative sequences of the clusters were then used for construction of sequence similarity network (SSN) from the website (http://efi.igb.illinois.edu/efi-est/stepa.php)[28]. The network parameters used were: (1) *E*-value, 5; (2) fraction, 1; and (3) alignment score, 6.

**Sequence analyses of CCHs.** The 1694 similar proteins of YtkR7 were obtained using YtkR7 (I3NN73) as query from the InterPro website (http://www.ebi.ac.uk/interpro/search/sequence-search)[20]. The 1694 sequences, YtkR7, and C10R6 were used for further sequence similarity network analysis of CCHs. The network parameters used were: (1) *E*-value, 5; (2) fraction, 1; and (3) alignment score, 38. The Cytoscape software was used to visualize the sequence similarity networks[28].

**Sequence logo analysis.** Multiple sequence alignment was executed using Clustal Omega on the website (http://www.ebi.ac.uk/Tools/msa/clustalo/), and the alignment results were used to create the sequence logo on the website (http://weblogo.threeplusone.com/create.cgi)[29].

**Protein expression and purification.** DNA isolation and manipulation in *E. coli* or *Streptomyces* strains were performed following standard methods[30]. PCR amplifications were carried out on a Thermal Cycler (Applied Biosystems). The PCR products were cloned into the pMD19-T vector cloning kit (Takara Biotechnology (Dalian) Co., Ltd), confirmed by DNA sequencing (Shanghai Biosune Biotech Co., Ltd (China)), digested with *Nde* I and *Hin* dIII/*Xho* I, and ligated into the final expression vector pET-28a(+) and pET-37b(+), respectively.

The resulting recombinant plasmids were introduced into *E. coli* BL21(DE3) for protein overexpression. When the cultures (400 mL) in LB media supplemented with kanamycin (50 μg mL$^{-1}$) were grown to an OD$_{600}$ of 0.6–1.0 at 37 °C, protein expression was induced by the addition of isopropyl-β-D-thiogalactopyranoside (IPTG) to a final concentration of 0.1 mM, followed by further incubation for 18–24 h at 16 °C. Then, the cultures were centrifuged for 10 min at 5000 rpm 4 °C. The *E. coli* cell pellet was resuspended in ~30 mL of lysis buffer (50 mM NaH$_2$PO$_4$, 500 mM NaCl, 10 mM imidazole, and 10% glycerol, pH 8.0), and the His-tagged proteins were purified by Ni-NTA affinity chromatography according to the manufacturer's manual (Qiagen). The eluted protein was desalted using a PD-10 Desalting Column (GE Healthcare, USA) and the purified protein was stored at −80 °C in buffer (50 mM NaH$_2$PO$_4$, 100–300 mM NaCl, 10% glycerol, pH 7.0) for further enzymatic assays. Bradford assays were used to determine the concentration of the purified proteins.

**Site-directed mutagenesis.** PCR amplifications (18–26 cycles) were carried out using *ytkR7* and *lin2189* expressing plasmids as template. After gel extraction (Generay Biotech (Shanghai) Co., Ltd), the PCR products were digested by *Dpn* I and the digested products were directly used to transform *E. coli* DH5α. Each point mutation was confirmed by DNA sequencing. The resulting site-mutated plasmids were introduced in *E. coli* BL21(DE3) according to the procedures described above for the wild-type protein expressions.

**Enzymatic assays.** The YTM (or CC-1065) hydrolysis reaction was carried out in 50 mM NaH$_2$PO$_4$ buffer containing 5–10 μM protein, 25% DMSO, and 100–300 μM YTM (or CC-1065) in a 30 μL system at pH 7.0. The reaction was incubated for 25–45 min at 25 °C, then quenched by mixing with 30 μL DMSO, and finally centrifuged for 5 min at 12,000 rpm. The supernatant was analyzed by high performance liquid chromatography (HPLC).

**Bacterial growth inhibition assays.** The *E. coli* BL21(DE3) strains transformed with recombinant plasmids were induced for expression of CCHs for about 5 h at 16 °C. The resulting cultures were adjusted to the same concentration, and then transferred to fresh LB media in test tube containing IPTG (0.1 mM), kanamycin (50 μg·mL$^{-1}$), and YTM (or compound **5**) with different concentrations to grow about 12–24 h, respectively. Their final *E. coli* cultures were diluted and the concentrations were measured by the absorbance at 600 nm, respectively.

When the plates were used, the adjusted cultures were mixed with 2% low melting temperature agarose containing LB media and then were poured on the LB plates, respectively. In total, 2 μL different concentrations of YTM were dropped on the plate surface for further incubation about 20–36 h. Two independent replicates were conducted for each assay.

**Cytotoxicity assay for YTM and compound 5.** All cell-culture work was conducted in a class II biological safety cabinet. Jurkat cells were maintained in RPMI-1640 medium supplemented with 10% fetal bovine serum (FBS). Jurkat cells were grown to approximately 80% confluence, and then collected, followed by centrifugation (3 min at 1000 rpm). The supernatant was discarded and the cell pellet was resuspended in fresh medium, and the concentration of cells was determined using a hemacytometer. The cell suspension was diluted to a concentration of 15,000 cells per 50 μL. The wells of a pre-sterilized 96-well plate were charged with 50 μL per well of the diluted cell suspension. Stock solutions of each compound in DMSO were diluted serially with RPMI-1640 medium (supplemented with FBS), and the resulting 50 μL solutions were added to the wells containing cells to achieve final concentrations of 0.1 nM–1.6 nM of YTM or 0.1 nM–10 μM of compound **5**. After incubating at 37 °C and in 5% CO$_2$ for 48 h, 10 μL of CCK-8 (Vazyme$^{TM}$ CCK-8 Cell Counting Kit) was added to each well. After incubating at 37 °C and in 5% CO$_2$ for another 4 h, the optical density at 450 nm was read out with a microplate reader and the background was subtracted at 650 nm. Cells incubated with 0.1% DMSO served as control. Three independent replicates were conducted.

**Construction and complementation of the *ytkR7* replacement mutant.** The *ytkR7* gene was inactivated by PCR-targeting method using the spectinomycin resistance cassette (*aadA*) (R7KO1 and R7KO2 as primers in Supplementary Table 1). To conduct complementation assays, the wild-type *ytkR7* or *c10R6* controlled by *ermE* promoter were cloned into the pSET152 plasmid. Conjugation from *E. coli* S17-1 to *Streptomyces* was performed according to our previous studies[9, 30]. Specifically, *E. coli* S17-1 strains (containing the recombinant plasmids) were gently mixed with the *Streptomyces* spores, and then the mixture was plated on the IWL-4 solid medium. After incubation at 30 °C for 12–16 h, the antibiotics spectinomycin (or apramycin) and nalidixic acid were added on the plates. Positive clones were identified 5 days later.

**Production and characterization of the relative metabolites.** Production of YTM and other relative metabolites was carried out using ISP-2 medium (yeast extract 4 g, malt extract 10 g, glucose 4 g, agar 20 g, and distilled H$_2$O to 1 L, pH 7.2)[9].

Production of CC-1065 and other relative metabolites by *Streptomyces zelensis* NRRL 11183 was performed using our optimized medium (dextrin 30 g, fish meal 10 g, cornmeal 30 g, cottonseed meal 30 g, glucose 10 g, Na citrate 2.5 g, MgSO$_4$•7H$_2$O 1 g, CaCO$_3$ 5 g, FeSO$_4$•7H$_2$O 0.02 g, KCl 0.5 g, CoCl$_2$•7H$_2$O 0.02 g, Na$_2$HPO$_4$•12H$_2$O 3 g, pH 7.0, and distilled H$_2$O to 1 L).

HPLC analysis was conducted on the Agilent 1200 HPLC system (Agilent Technologies Inc., USA) with a reverse-phase Alltima C18 column (5 μm, 4.6 × 250 mm). The conditions include: Solvent A was H$_2$O and Solvent B was CH$_3$CN; the gradient program was 0–3 min 15% B, 3–6 min 15–40% B, 6–12 min 40% B, 12–19 min 40–55% B, 19–22 min 55–85% B, 22–28 min 85% B, and 28–29 min 15% B; the flow rate was 1 mL min$^{-1}$ with DAD detector.

The crude extracts of fermentation were subjected to the reverse-phase silica gel column chromatography. All the fractions were monitored by HPLC. The desired fractions were further purified by semi-preparative HPLC on a Shimadzu LC-20-AT system. High-resolution ESI-MS analysis was conducted on the 6230B Accurate Mass TOF LC/MS System (Agilent Technologies Inc., USA). NMR data were recorded on the Bruker Avance III AV400 (Cryo) spectrometers (Bruker Corporation Co., ltd, Germany) and on the Agilent ProPlus 500 MHz NMR spectrometer (Agilent Technologies Inc., USA).

**Surface plasmon resonance experiments.** All experiments were performed in PBS buffer pH 7.4 (NaCl 137 mM, KCl 2.7 mM, Na$_2$HPO$_4$ 10 mM, KH$_2$PO$_4$ 2 mM, 5% DMSO, and 0.05% Tween 20) on a Biacore T200 System at a flow rate of 30 μL min$^{-1}$ and 25 °C. YtkR7, Rob, and SbmC proteins were immobilized on a CM5 sensor chip at 23,757.5, 14,397.3, and 3827.0 response units (RU), respectively.

**Crystallization and data collection.** The lin2189 mutant proteins were purified by one Ni-NTA column and one size exclusion column (Superdex 200 16/600, GE Health Care). Crystals with high qualified diffraction for data collection were first obtained using lin2189 mutant E157A/E185L, which lost the activity of YTM hydrolysis. A mixed solution contains 10% 15 mM YTM solved in DMSO and 1

mM protein was used for incubation and then for crystal growth by sitting drop vapor diffusion method at 16 °C. The crystals were further soaked in a reservoir solution containing 10% glycol and 5% 15 mM YTM before being flash frozen in liquid nitrogen. Data were collected at BL19U1 beamline in Shanghai Synchrotron Radiation Facility at the wavelength of 0.97791 Å and processed with the HKL3000 package.

**Structure determination and refinement**. The complex structure of lin2189 E157A E185L-YTM was determined by molecular replacement using search model PDB ID 3B49. YTM was added into the structure based on the omit Fo–Fc map and iterative cycles of refinement were carried out using COOT, Refmac, and PHENIX. A random selection of 5% reflections was set aside for cross-validation. PROCHECK and MolProbity were used to access the overall quality of the structural models. All statistics for data collection and structural refinement were listed in Supplementary Table 3. Structure figures were made using PyMol 1.3.

**Loop filling and refinements**. Since the loop B was missing in the complex structure, loop refinement protocol in Modeller V9.10[31] was used to fulfill and refine this area. A total of 20,000 loops were generated and a conformation with the lowest DOPE (Discrete Optimized Protein Energy) score was chosen for receptor construction.

**Protein structure preparations**. All protein models were prepared in Schrodinger suite software under OPLS_2005 force field[32]. Hydrogen atoms were added to repaired crystal structures according to the physiological pH (7.5) with the PROPKA tool[33] in Protein Preparation tool in Maestro[34] to optimize the hydrogen bond network. Constrained energy minimizations were conducted on the full-atomic models, with heavy atom coverage to 0.4 Å.

**Ligand structure preparations**. All ligand structures were produced in Schrodinger Maestro software[34]. The LigPrep module in Schrodinger software was introduced for geometric optimization by using OPLS_2005 force field. The ionization states of ligands were calculated with Epik tool[35] employing Hammett and Taft methods in conjunction with ionization and tautomerization tools[35].

**Protein–ligand docking**. The docking of a ligand to the receptor was performed using Glide[36]. Cubic boxes centered on the ligand mass center with a radius 10 Å for all ligands defined the docking binding regions. Flexible ligand docking was executed for all structures. Twenty poses per ligand out of 20,000 were included in the post-docking energy minimization. The best scored pose for the ligand was chosen as the initial structure for MD simulations.

**Molecular dynamics simulations**. All unbiased MD simulations were performed in Gromacs 5.1.4[37]. All amino acid residues of the protein were modeled according to their protonation state at neutral pH. The protein was centered in a water box with a distance of 16 Å away from the protein. The total number of atoms was approximately 56,000: 54 Na$^+$ and 52 Cl$^-$ ions, and about 18,000 water molecules. Amber99SB*-ILDNP[38] force field was assigned to the protein, water, and ions, while the ligands were treated by AmberGAFF2 force field[39] through ACPYPE tool[40]. The ligands were submitted to GAUSSIAN 09 program[41] for structure optimization at Hartree-Fock 6–31 G* level prior to the generation of force field parameters. All bond lengths of hydrogen atoms in the system were constrained using M-SHAKE[42]. Van der Waals and short-range electrostatic interactions were cut off at 10 Å. The whole system was heated linearly at constant volume (NVT ensemble) from 0 to 310 K over 400 ps. Ten nanoseconds equilibration was performed at constant pressure and temperature (NPT ensemble; 310 K, 1 bar) using the Nose–Hoover coupling scheme with two temperature groups. Long-range electrostatic interactions were computed by particle mesh Ewald (PME) summation. Finally, 400 ns MD simulations with a time step of 2.0 fs were performed for apo lin2189, lin2189-YTM, and lin2189-product. The MD simulations results were analyzed in Gromacs[37] and VMD[43]. Figures were prepared in PyMOL and Inkscape[44]. The pKa calculation of both E185 and E157 was done in PROPKA tool[33]. We extracted 500 structures through the APO MD simulation and used PROPKA[33] for calculation. The PCA analysis was performed on the nonbiased MD simulation of APO protein.

**Metadynamics simulations**. Free-energy profiles of the systems were calculated by using well-tempered metadynamics in Gromacs 5.1.4[37] with Plumed 2.1.2[45] patches. Metadynamics adds a history-dependent potential $V(s, t)$ to accelerate sampling of the specific CVs $s(s_1, s_2, …, s_m)$[46]. $V(s, t)$ is usually constructed as the sum of multiple Gaussians centered along the trajectory of the CVs (Eq. 1).

$$V(s,t) = \sum_{j=1}^{n} G(s, t_j) = \sum_{j=1}^{n} w_j \prod_{i=1}^{m} \exp\left(-\frac{[s_i(t) - s_i(t_j)]^2}{2\sigma^2}\right) \quad (1)$$

$T$ is the temperature of the system. The CVs sample an ensemble at a temperature $T + \Delta T$, which is higher than the system temperature $T$. Periodically, during the simulation, another Gaussian potential, whose location is dictated by the current values of the CVs, is added to $V(s, t)$[47]. In our simulations, the mass center distances between ligands (substrate or product) and E185, and that between loop B and loop C were assigned as two CVs $s_1$ and $s_2$, while the width of Gaussians, $\sigma$, was set to 0.5 Å. The time interval, $\tau$ was 0.09 ps. Well-tempered metadynamics involves adjusting the height, $w_j$, in a manner that depended on $V(s, t)$ where the initial height of Gaussians $w$ was 0.5 kcal mol$^{-1}$, the simulation temperature was 310 K, and the sampling temperature $\Delta T$ was 298 K. The convergence of our simulations was judged by using the free energy difference between states X and A at 10 ns intervals. State X is a reference state, while A is the state after additional biases added to state X. Once the resulting data remained stable over time, the simulation was considered as converged. Each metadynamics simulation lasted 120 ns, and the results were analyzed upon convergence.

**Interaction fingerprint calculations**. The interaction fingerprint between protein and ligand was done with PLIP tool[48]. We first extracted snapshots of the final 20 ns MD simulation. We then used PLIP to convert protein–ligand coordinates into a bit-string fingerprint (TIFP) registering the corresponding molecular interaction pattern. TIFP fingerprints have been calculated for 500 protein–ligand complexes for each case, enabling a broad comparison of relationships between interaction pattern similarities and ligand or binding site pairwise similarities[49]. In this work we kept the default parameters of PLIP and focused on three types of interactions: H-bond interaction, π stacking and hydrophobic contact.

**Data availability**. The atomic coordinates and structural factors of lin2189 E157A/E185L and lin2189 E157A/E185L-YTM have been deposited in the Protein Data Bank under accession codes 5 × 5 R and 5 × 5 M, respectively. The data that support the findings of this study are available from the corresponding author upon request.

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

## Acknowledgements

We thank Prof. Zixin Deng's Lab at Shanghai Jiao Tong University for technical assistance in MS/MS data collection and Dr. Jianrong Xu at Shanghai Jiao Tong University for surface plasmon resonance experiments. We also thank the staff at beamline BL17U1 of the Shanghai Synchrotron Radiation Facility (China) and beamlines BL19U1 and 18U2 of National Center for Protein Science Shanghai (China) for access and help with crystal data collection. This work was supported in part by grants from NSFC (21632007, 21502217, and 21621002), STCSM (15ZR1449400 and 15JC1400400), CAS (XDB20000000 and QYZDJ-SSW-SLH037), and State Key Laboratory of Microbial Metabolism, Shanghai Jiao Tong University (MMLKF15-02). The molecular modeling and molecular dynamics simulation were performed at the Interdisciplinary Centre for Mathematical and Computational Modeling in Warsaw (GB70-3 and GA65-23).

## Author contributions

G.-L.T. and H.Y. conceived the study and designed the experiments. H.Y. and W.H. participated in the genetic experiments. H.Y. performed the biochemical assays. H.Y., S.W., K.Y., W.-B.J. and Y.L. isolated the compounds and characterized their structures. J. Zhang and Y.C. determined crystal structures. S.C. and S.Y. performed the molecular modeling and molecular dynamics simulations. H.Y. and Y.Y. carried out the cytotoxic assays. Y.I. provided the strain *Streptomyces* sp. TP-A0356. H.Y., S.Y., J. Zhou and G.-L. T. interpreted the data and wrote the manuscript.

## Additional information

**Competing interests:** The authors declare no competing financial interests.

