## [Peer Review File · Nature Communications]

Reviewers' comments:

Reviewer #1 (Remarks to the Author):

The authors are the first to report convincing evidence of cyclopropanoid cyclopropyl hydrolase activity by a subfamily of GyrI-like proteins. The study focuses partially on YtkR7, the gene product of a GyrI-like protein encoded within the same biosynthetic gene cluster of Yatakemycin, a cyclopropanoid antitumor antibiotic isolated initially from *Streptomyces* sp. TP-A0356. Importantly, such an activity is potentially relevant to mechanisms of resistance to DNA-alkylating agents such as Yatakemycin. Moreover, this work suggests resistance related functions for single-domain GyrI-like proteins whose biological functions remain largely uncharacterized. Findings like those reported here suggest a biological importance of GyrI-like protein functions, which are wide-spread in all prokaryotes, eukaryotes, and archaea. Currently, their functions remain unknown. As such, the discovery of the described functions by GyrI-like proteins is indeed novel and noteworthy.

Authors provide convincing genetic, biochemical and additional bioinformatic-related evidence that YtkR7 is responsible for the metabolism and inactivation of 1 (YTM) and that other related GyrI-like proteins catalyze similar reactions. Importantly, the authors' expertise in the activity of Yatakemycin and related compounds, as well as other protein biosynthetic gene clusters, is well-established. The review is confident in the biochemical aspects of the work.

However, the discussions and analyses of structural aspects of the observed cyclopropanoid cyclopropyl hydrolase are unclear and, in their current state, contribute weakly to the manuscript. Whereas the reviewer feels that major changes (and corrections) are needed (see below), a revised manuscript should be required for acceptance.

Major Concerns:

If indeed YtkR7 functions as a cyclopropanoid cyclopropyl hydrolase, the authors should show that the protein shows significant rate acceleration over the background reaction, especially given the reactivity of strained cyclopropane rings. Also, to better demonstrate YtkR7's enzymatic properties and power as a catalyst, the authors may want to show a Michaelis-Menton analysis and determine the kinetic parameters. Importantly, quantitative insights regarding catalysis might be important for determining or establishing the contributions to cellular resistance to compounds like YTM.

Along these lines, does the bacterial strain containing the YtkR7 knockout display a phenotype (resulting from the build up of YTM, from which YtkR7 is presumably providing protection against)?

The authors state in the main text and show in Supplementary Figure 8 that both DMSO and low pH increase YtkR7-mediated hydrolysis of YTM. Did the authors determine if the background (uncatalyzed) reaction is also promoted under similar conditions? If the reaction does not occur in the absence of YtkR7 or the other GyrI-like proteins, under any conditions, the authors should state this explicitly.

To further probe the catalytic mechanism of the GyrI-like CCHs, the authors solve the X-ray structure of a catalytically deficient variant of lin2189 with 1. The authors do not clearly show the interactions between lin2189 and 1. Also, the figures, including the supplementary information, does not include any electron density maps for the reviewer and readers to judge the quality of their model or clearly visualize the protein-ligand interactions. At the very least, the authors should show the ligand electron density as well as residue side chains and water molecules involved in binding.

Because the observed ligand-binding mode likely does not represent a catalytically relevant one, a docking approach is used to model the latter. However, there is no clear representation of the modeled ligand-binding mode or an indication of how it differs from that observed by crystallography. (The mutations introduced into the crystallized variant might be the cause of the observed, alternative binding mode.) Whereas the docked mode is described in the text by the authors, interpretation should be left to the reader via a clear visual presentation of the binding pocket, including residues making contact with the ligand. Also, the authors offer no evaluation of the predicted docking mode. They do state that “YTM can be docked well into the lin2189 pocket.” (Supplementary Figure 17) This statement means nothing. Finally, do the authors observe significant structural differences between the apo- and ligand-bound lin2189 binding modes?

Importantly, the shapes of YTM and CC-1065 also may be conducive to binding modes that flip the 1L and 1R portions of the molecules. Did the authors consider this in their ligand-modeling?

It is not clear how the MD simulations validate or were even motivated by the binding mode suggested by docking. The authors should state such motivations explicitly, making it easier for the reader to appreciate the results. Also, are the observed loop motions required to obtain catalytically relevant ligand-binding modes or explain broad ligand specificity? Do the X-ray structures support that loops A, B and C are flexible as the authors claim.

The authors state, "Notably, the complex crystal structure is identical to snapshots from our

metaMD simulations, which further confirms our identification of this substrate entrance." This statement is unclear. Moreover, the MD simulations cannot confirm such an entrance.

The authors state, "In apo-CCHs, E157 is protonated and E185 is deprotonated, which has been confirmed by pKa calculations on the MD trajectory (Supplementary Fig. 22)." MD simulations cannot confirm pKa values. However, the apo-lin2189 structure may support such an assignment. Both E185 and E157 are somewhat buried and appears to engage with each other via an H-bonding contact.

The authors may want to consider moving Figure 4 and the discussion of the presented results to an earlier section in the paper.

Other Issues:

Figure 1 legend does not explain the IC50 values shown for compounds 1, 2, 3 and 4.

Figure 2 should present the lin2189 structure in same orientation and size for viewing clarity and analysis of the results.

Figure 3. The proton transfer shown in part d appears to be improbable considering the relative locations of E157 and E185 as well as the structure of the substrate.

The color differences corresponding to sequence similarity in Supplementary Figure 9 are not clear.

Supplementary Figure 15. One interesting aspect of the lin2189 structure is how differs from known GyrI-like proteins. This topology figure presents a good opportunity to show this difference.

The title of Supplementary Figure 16 is, "The recognition patterns of substrate molecules in an asymmetric unit were consistent in the crystallization environment." It is not clear what this figure is supposed to convey.

Supplementary Figure 19 is a very good figure. Space permitting, the authors may want to move it to the main text.

The bar-graph in Part C of Supplementary Figure 20 (Site-directed mutagenesis experiments of CCHs using YTM as substrate) has no label or units.

The legend of Supplementary Figure 22 should read, "The calculated pKa values of E157 and E185 for apo-lin2189, lin2189-YTM and lin2189-5". The standard error is indicated by an error

bar.

Reviewer #2 (Remarks to the Author):

In this work, the Authors present a characterization of a particular subfamily of the GyrI-like proteins, and examine possible reasons leading to resistance, related to YTM cyclopropyl hydrolysis. The topic seems relevant, however, the way it is presented is too condensed and cryptic to permit the reader to get familiar with the problem and to have a clear picture of the gap filled by this work.

To probe the catalytic mechanism of CCHs, the Authors carried out both experimental and computational work on a member of CCH subfamily. In order to get a crystal, they made two mutations impairing the catalytic activity. In this way, they are no longer aiming at probing the catalytic mechanism, but just the binding one. However, I would say these mutations may also affect the binding properties of the protein, since they, for instance, affect the local electrostatic potential in a quite dramatic way. The Authors should have expected that these mutations very likely affect binding and should comment on this.

Then, in order to increase the sampling of the binding event the Authors used an enhanced sampling technique, namely metadynamics. Then, by using Principal (not principle) Component analysis, they observe a high fluctuation of the loop B and C distance. However, this quantity is one of the two collective variables accelerated by the metadynamics method, so how reliable can be this observation? Also the subsequent observations are flawed by the same issue. The correct usage of metadynamics would be to wait for convergence, which on a 2D space could be not straightforward, and to derive considerations on the reconstructed free energy surface, rather than commenting on individual events observed in a biased trajectory and ascribing them to a possible induced fit effect.

In the main text, they say that "In apo CCHs, E157 is protonated and E185 is deprotonated, which has been confirmed by pKa calculations on the MD trajectory", but in the Methods they say that "All amino acid residues of the protein were modelled according to their protonation state at neutral pH", so how do they estimate pKa of titratable residues in the MD runs? How therefore are derived the data in Supp. figure 22?

The protein was put in a water box 16Å away from the protein, but this distance should be monitored during the dynamics, unless they put some restraints on translation or rotation of the system.

"(NVT ensemble, 1bar)" is evidently a mistake.

"the simulation temperature was 310 K, and the sampling temperature ΔT was 298 K" please clarify.

"The convergence of our simulations was judged by using the free energy difference between states X and A at 10 ns intervals. Once the resulting data remained stable over time, the simulation was considered as converged. Each metadynamics simulation lasted 120 ns, and the results were analyzed upon convergence." Many things in this sentence should be rephrased and clarified, also because convergence is probably the most delicate aspect of metadynamics, as in many other enhanced sampling approaches. How precisely are defined X and A? how many recrossings have been observed? which is the threshold used for convergence? etc..

Overall, I should recommend a much clearer presentation of the logical flow of the work and a more careful and appropriate usage of the computational tools.

Reviewer #3 (Remarks to the Author):

GyrI-like proteins have been reported to confer antibiotic resistance through binding of small molecules. In this report, Yuan and co-workers report the catalytic function of GyrI-like proteins that hydrolyzes the cyclopropyl ring of cytotoxic cyclopropanoids. The authors initially identified this catalytic function through gene deletion study, which revealed the disappearance of two hydrolysis product peaks in the ytkR7 gene knock out mutants. Subsequent characterizations of YtkR7 homologues showed that many of the GyrI-like proteins have the same capability to hydrolyze cyclopropanoids. In addition, the authors conducted structural characterization of lin2189, which is a YtkR7 homolog that can hydrolyze multiple cyclopropanoid substrates. While the apo structure of this protein has been previously reported, the authors succeeded in acquiring the co-crystal of 1 with catalytically inactive mutant lin2189. Protein-ligand docking was used to predict the binding of 1 and 2 to lin2189 in the catalytically relevant form. In addition, metaMD simulations were performed to observe the potential movement in loops B and C for substrate entrance and exit. Finally, the authors confirmed that the cytotoxicity of the hydrolyzed product is much lower than the cyclopropanoids and the growth of *E. coli* in presence of 1 is improved upon expression of the cyclopropanoid cyclopropyl hydrolases. Finally, SPR analysis was performed to show that three GyrI-like proteins, SbmC, Rob and YtkR7, are capable of binding antibiotics as one of the detoxification methods. Overall, the work presented is clear and well rounded. Although the structure of the protein has been reported previously, additional computational modeling revealed additional information regarding the catalytic properties of the GyrI-like proteins. Based on the importance of the GyrI-like protein in antibiotic resistance, the reviewer judges that this work is of interest to

broad audience and is suitable for publication in Nature Communications with minor revision.

1. Because this is the first example for the catalysis of GyrI proteins, it is informative if the authors provide the kinetic values (K_m and k_{cat}) of one of the identified cyclopropyl ring hydrolases. The enzymatic traits such as pH and temperature dependency should be also presented.
2. Can the authors explain how they find the key acidic residues E157 and E185 before crystallization briefly?
3. The authors report that 1694 GyrI proteins that has similar catalytic residues as lin2189. Are the genes encoding them located close to yatakemycin gene cluster? Can the authors predict the substrate for these enzymes from the point of view of the bioinformatic aspect?
4. How do the cyclopropyl ring hydrolases bind with the antibiotics except yatakemycin? Do they also bind with the substrate-binding cage? The authors should explain the difference of the binding mode between the cyclopropyl ring hydrolases and the reported GyrI protein structures.
5. Last sentence on page 5 seems to imply that substrates other than 1 and 2 were used for the in vitro and in vivo studies, but it is unclear which molecules were actually tested.
6. Did the authors observe the formation of the spontaneously oxidized product of 7?
7. Isolated compounds do not appear to be entirely pure based on the NMR spectra provided.
8. The HPLC peak intensities of the assays comparing the activities of the mutants are different for the double mutant compared to the other single mutants. Is there any reason for this difference?

The following are our point-by-point responses.

Reviewer #1 (Remarks to the Author):

The authors are the first to report convincing evidence of cyclopropanoid cyclopropyl hydrolase activity by a subfamily of GyrI-like proteins. The study focuses partially on YtkR7, the gene product of a GyrI-like protein encoded within the same biosynthetic gene cluster of Yatakemycin, a cyclopropanoid antitumor antibiotic isolated initially from *Streptomyces* sp. TP-A0356. Importantly, such an activity is potentially relevant to mechanisms of resistance to DNA-alkylating agents such as Yatakemycin. Moreover, this work suggests resistance related functions for single-domain GyrI-like proteins whose biological functions remain largely uncharacterized. Findings like those reported here suggest a biological importance of GyrI-like protein functions, which are wide-spread in all prokaryotes, eukaryotes, and archaea. Currently, their functions remain unknown. As such, the discovery of the described functions by GyrI-like proteins is indeed novel and noteworthy.

Authors provide convincing genetic, biochemical and additional bioinformatic-related evidence that YtkR7 is responsible for the metabolism and inactivation of 1 (YTM) and that other related GyrI-like proteins catalyze similar reactions. Importantly, the authors' expertise in the activity of Yatakemycin and related compounds, as well as other protein biosynthetic gene clusters, is well-established. The review is confident in the biochemical aspects of the work.

However, the discussions and analyses of structural aspects of the observed cyclopropanoid cyclopropyl hydrolase are unclear and, in their current state, contribute weakly to the manuscript. Whereas the reviewer feels that major changes (and corrections) are needed (see below), a revised manuscript should be required for acceptance.

Major Concerns:

If indeed YtkR7 functions as a cyclopropanoid cyclopropyl hydrolase, the authors should show that the protein shows significant rate acceleration over the background reaction, especially given the reactivity of strained cyclopropane rings.

Response: Sorry for the misunderstanding. In fact, YTM is very stable under all our reaction conditions. We also did the control assays. For example, in Figure 1d(i), the control assay contained only the reaction buffer and the substrate YTM. We have modified the corresponding legends. In Figure 1d(ii), the reason for the trace amount of 5 production is that YtkR7 was not completely inactivated by boiling.

Also, to better demonstrate YtkR7's enzymatic properties and power as a catalyst, the authors may want to show a Michaelis-Menton analysis and determine the kinetic parameters.

Response: Because the substrates (yatakemycin and CC-1065) are difficult to dissolve in the reaction buffer, we need to add DMSO (25% V/V, final) into the reaction system. Despite many efforts, we failed to obtain satisfactory values for V_{max} and K_M . For example, we obtained the following fitting curves.

Importantly, quantitative insights regarding catalysis might be important for determining or establishing the contributions to cellular resistance to compounds like YTM.

Response: From the HPLC analyses of the fermentation products, we can estimate that the yield of yatakemycin in the *ytkR7* mutant is about one third that of the wild type.

Along these lines, does the bacterial strain containing the YtkR7 knockout display a phenotype (resulting from the build up of YTM, from which YtkR7 is presumably providing protection against)?

Response: Factually, the *ytkR7* knockout strain displays a bald phenotype, that is, the spore production is very scarce. The phenotype is restored normally when we perform the complementation using the native *ytkR7*. However, we also observed the bald phenotype in

other gene knockout mutants that could not produce YTM. So we cannot correlate the phenotype with YTM production.

The authors state in the main text and show in Supplementary Figure 8 that both DMSO and low pH increase YtkR7-mediated hydrolysis of YTM. Did the authors determine if the background (uncatalyzed) reaction is also promoted under similar conditions? If the reaction does not occur in the absence of YtkR7 or the other GyrI-like proteins, under any conditions, the authors should state this explicitly.

***Response:* Sorry for the misunderstanding. We usually make a stock solution of YTM in DMSO. As mentioned above, YTM is very stable under all our reaction conditions. We also did the control assays that contained only the reaction buffer and the substrate. As stated in the text “As expected, incubation of YtkR7 with 1 afforded 5, which was not observed for the control assay (Fig. 1d).”, the underlined words have explained that the enzyme YtkR7 is absolutely necessary. We have also modified the corresponding legends.**

To further probe the catalytic mechanism of the GyrI-like CCHs, the authors solve the X-ray structure of a catalytically deficient variant of lin2189 with 1. The authors do not clearly show the interactions between lin2189 and 1.

***Response:* We have also discussed this point with all authors intensively. Initially we tried to illustrate the binding mode presenting all side chains of related amino acids. Unfortunately, the ligand itself is very big and there are so many residues involved in the interactions that everything is stacking together if we show all side chains. It is very difficult to see the details clearly and also there is not enough space for labeling. We prepared the following figure showing all residue side chains as an example. That’s why we introduced this simplified diagram (Figure 2a & 2b) coupled with interaction finger print (IFP) plot (Figure 2c). To further make this point clearly, we provide another 2D interaction diagram as Supplementary Figure 16.**

Also, the figures, including the supplementary information, does not include any electron density maps for the reviewer and readers to judge the quality of their model or clearly visualize the protein-ligand interactions. At the very least, the authors should show the ligand electron density as well as residue side chains and water molecules involved in binding.

Response: We thank the reviewer's suggestions. We provide the electron density map of the ligand as Supplementary Figure 17.

Because the observed ligand-binding mode likely does not represent a catalytically relevant one, a docking approach is used to model the latter. However, there is no clear representation of the modeled ligand-binding mode or an indication of how it differs from that observed by crystallography. (The mutations introduced into the crystallized variant might be the cause of the observed, alternative binding mode.) Whereas the docked mode is described in the text by the authors, interpretation should be left to the reader via a clear visual presentation of the binding pocket, including residues making contact with the ligand.

Response: We have showed the binding modes of both ligands in crystal structure (Figure 2a) and from docking results (Figure 2b) side by side. As explained above, there are too many residues to see the interaction clearly if we present the side chain explicitly. However, we added another 2D diagram as Supplementary Figure 16 for this.

Also, the authors offer no evaluation of the predicted docking mode. They do state that “YTM can be docked well into the lin2189 pocket.” (Supplementary Figure 17) This statement means nothing.

Response: In the Figure, we have shown all residues involved in the ligand CC-1065 interaction as spheres. Most of the residues have now been proved playing important roles by our biochemical mutations as shown in Supplementary Fig. 20.

Finally, do the authors observe significant structural differences between the apo- and ligand-bound lin2189 binding modes?

Response: Yes, we observed noticeable differences between the apo state and ligand-bound crystal structure, especially in loop A, B and C. Both loop B and C are missing in the apo structure due to their flexibility, whereas loop A underwent noticeable shifting. Here is a figure for this point.

Importantly, the shapes of YTM and CC-1065 also may be conducive to binding modes that flip the 1L and 1R portions of the molecules. Did the authors consider this in their ligand-modeling?

Response: We thank the reviewer mentioning this point. We do have considered this point. In our docking, all the top 5 ranked poses are identical and we didn't obtain the 1L/1R flipped binding mode in top scored poses. 1L/1R flipped binding mode has been observed in poorly scored pose. The docking energy with our proposed binding mode and 1L/1R flipping pose is over 5 kcal/mol, indicated that our proposed binding mode is much more favorable than 1L/1R mode. More importantly, our crystal structure density map (Supplementary Fig. 17) also revealed identical relative position of 1L and 1R.

It is not clear how the MD simulations validate or were even motivated by the binding mode suggested by docking. The authors should state such motivations explicitly, making it easier for the reader to appreciate the results. Also, are the observed loop motions required to obtain catalytically relevant ligand-binding modes or explain broad ligand specificity? Do the X-ray structures support that loops A, B and C are flexible as the authors claim.

Response: Induced fit effect (IFE) has been frequently observed in many biological systems, including enzyme catalysis, when a ligand binds to a protein inducing noticeable motions of the protein. This has been discussed in the main text. It has also been extensively discussed in the following highly cited papers.

- (1) Sherman, W. et al. Novel procedure for modeling ligand/receptor induced fit effects (2006) *J Med Chem*, 49:534-553. (citation=897)
- (2) Karpowich, N. et al. Crystal structures of the MJ1267 ATP binding cassette reveal an induced-fit effect at the ATPase active site of an ABC transporter (2001) *Structure*, 9:571-586. (citation=305)
- (3) Hammes, G.G. et al. Conformational selection or induced fit: a flux description of reaction mechanism (2009) *Proc Natl Acad Sci U S A*, 106:13737-13741. (citation=289)

Our MD simulation reveals that loop A, B and C underwent noticeable motion which is in excellent agreement with our crystal structures as discussed above.

The authors state, "Notably, the complex crystal structure is identical to snapshots from our metaMD simulations, which further confirms our identification of this substrate entrance." This statement is unclear. Moreover, the MD simulations cannot confirm such an entrance.

***Response:* We thank the reviewer's comments, and we rewrote this sentence in a more precisely way.**

The authors state, "In apo-CCHs, E157 is protonated and E185 is deprotonated, which has been confirmed by pKa calculations on the MD trajectory (Supplementary Fig. 22)." MD simulations cannot confirm pKa values. However, the apo-lin2189 structure may support such an assignment. Both E185 and E157 are somewhat buried and appears to engage with each other via an H-bonding contact.

***Response:* We thank the reviewer's suggestions. MD simulation itself cannot confirm the pKa value. However, we have calculated the pKa values on 500 structures extracted from MD simulations so that it is more reliable statistically. We add this point in the Method part.**

The authors may want to consider moving Figure 4 and the discussion of the presented results to an earlier section in the paper.

***Response:* CCHs are the first GyrI-like proteins disclosed as directly involved in antibiotic resistance, so we think it is significant to keep the figure in main text. We have added a discussion part to discuss our results.**

Other Issues:

Figure 1 legend does not explain the IC50 values shown for compounds 1, 2, 3 and 4.

***Response:* Sorry for our carelessness. We added the information in Figure 1a.**

Figure 2 should present the lin2189 structure in same orientation and size for viewing clarity and analysis of the results.

***Response:* We thank the reviewer's comments. Figure 2a,b show the details of the binding pocket and it requires zoomed-in position to visualize, whereas 2d,e,f only show the overall structure. The orientation of each panel is the same.**

Figure 3. The proton transfer shown in part d appears to be improbable considering the relative locations of E157 and E185 as well as the structure of the substrate.

***Response:* We thank the reviewer pointing this out. The distance seems to be a little bit far. Possibly the proton comes from water instead of E157. Considering E157 and E185 are much closer to each other, proton transfer in Figure 3e is more likely from E157. We correct this detail in Figure 3.**

The color differences corresponding to sequence similarity in Supplementary Figure 9 are not clear.

***Response:* We have included the identity and E-value for each protein in the figure legends, so the color differences might be accessory.**

Supplementary Figure 15. One interesting aspect of the lin2189 structure is how differs from known GyrI-like proteins. This topology figure presents a good opportunity to show this difference.

***Response:* We thank the reviewer's suggestion. We have provided structural superposition of BmrR, SbmC, and Rob as Supplementary Figure 27 for comparison. We also added discussion in the main text.**

The title of Supplementary Figure 16 is, "The recognition patterns of substrate molecules in an asymmetric unit were consistent in the crystallization environment." It is not clear what this figure is supposed to convey.

***Response:* We agree with the reviewer and we have removed this figure from the Supplementary Information file.**

Supplementary Figure 19 is a very good figure. Space permitting, the authors may want to move it to the main text.

***Response:* We thank the reviewer's suggestions. We have integrated the Supplementary Figure 19 into the Figure 2 in the main text.**

The bar-graph in Part C of Supplementary Figure 20 (Site-directed mutagenesis experiments of CCHs using YTM as substrate) has no label or units.

Response: Sorry for our carelessness. The Y axis indicates the relative capacity of 5 production. We have added the information.

The legend of Supplementary Figure 22 should read, “The calculated pKa values of E157 and E185 for apo-lin2189, lin2189-YTM and lin2189-5”. The standard error is indicated by an error bar.

Response: We changed this accordingly.

Reviewer #2 (Remarks to the Author):

In this work, the Authors present a characterization of a particular subfamily of the GyrI-like proteins, and examine possible reasons leading to resistance, related to YTM cyclopropyl hydrolysis. The topic seems relevant, however, the way it is presented is too condensed and cryptic to permit the reader to get familiar with the problem and to have a clear picture of the gap filled by this work.

To probe the catalytic mechanism of CCHs, the Authors carried out both experimental and computational work on a member of CCH subfamily. In order to get a crystal, they made two mutations impairing the catalytic activity. In this way, they are no longer aiming at probing the catalytic mechanism, but just the binding one. However, I would say these mutations may also affect the binding properties of the protein, since they, for instance, affect the local electrostatic potential in a quite dramatic way. The Authors should have expected that these mutations very likely affect binding and should comment on this.

Response: We thank the reviewer’s comments. Indeed, we would like to have minimum influence on the protein and close to the native state. The native enzyme will hydrolyze the substrate. To obtain enzyme-substrate complex crystal structure, mutation around the binding pocket is unavoidable, which leads to some changes of the protein.

The principles that we chose the key residues are as follows:

In 2004, Anantharaman *et al.* defined the GyrI-like family proteins. Based on the sequence of the MerR-like transcription activator BmrR, they found that the residue E185 (for lin2189) is highly conserved. [Anantharaman, V. and Aravind, L. (2004) *Proteins*, 56: 795-807.]

lin2189 E185 corresponds to BmrR E253. The BmrR E253 influences the ligand binding spectrum. Moreover, *in vitro* transcription data show that E253A and E253Q are constitutively active. [Newberry, K. J., et al. (2008) *J Biol Chem*, 283: 26795-26804.]

We thus mutated lin2189 E185, and found that it lost the enzymatic activity. Subsequently, we screened lin2189 E185L-YTM complex, and found that the electron density map of YTM is not clear enough. But from the information gained from the complex structure, we designed the E157 and E185 double mutant.

To make it clear, we added “Previous structural and biochemical studies of BmrR revealed that the glutamate residue Glu-253 has a noticeable effect on its ligand binding and transcription activation¹⁷. This acidic residue is highly conserved among the GyrI-like proteins¹¹, we thus constructed a lin2189 variant (E185L). Interestingly, this variant lost the enzymatic activity and allowed us to obtain a complex structure of lin2189 E185L and 1 in which the electron density for the substrate is partially visible. Detailed investigation of the electron density map implicated that several residues in the solvent accessible channel (e.g., E157) potentially interfered with substrate binding. Therefore, we designed a series of variants based on E185L. As expected, we successfully solved the complex structure of a catalytically-deficient form lin2189 E157A E185L and 1 with complete density at a resolution of 1.2 Å.” to the main text.

Then, in order to increase the sampling of the binding event the Authors used an enhanced sampling technique, namely metadynamics. Then, by using Principal (not principle) Component analysis, they observe a high fluctuation of the loop B and C distance. However, this quantity is one of the two collective variables accelerated by the metadynamics method, so how reliable can be this observation?

Response: We first would like to thank the review pointing this typo and we corrected it in the main text.

The PCA analysis is a widely used and well recognized method in biological study which is featured by the following frequently cited papers.

- (1) Haider, S. et al. *Molecular dynamics and principal components analysis of human telomeric quadruplex multimers* (2008) *Biophys J*, 95:296-311. (citation=132)
- (2) Balsera, M. A. et al. *Principal component analysis and long time protein dynamics* (1996) *J Phys Chem*, 100:2567–2572. (citaton=323)
- (3) David, C. C. et al. *Principal component analysis: a method for determining the essential dynamics of proteins* (2014) *Methods Mol Biol*, 1084:193-226. (citation=40)

PCA analysis has been proved to reflect the physiological properties of biological system. In our work, both loop B and C are missing in the crystal structure due to their high

flexibilities. Our PCA analysis on MD simulations is perfectly in agreement with this observation.

MetaMD is also a sophisticated and reliable method for rare biological event sampling. It can accelerate the rare event in a reasonable time scale. We have applied MetaMD in one of our previous work:

(1) Yuan, S. et al. *Activation of G-protein-coupled receptors correlates with the formation of a continuous internal water pathway* (2014) *Nat Commun*, 5:4733 (citation=57)

MetaMD has also been widely used in many other various biological systems which is represented by the following highly cited papers:

(1) Ensing, B. et al. *Metadynamics as a tool for exploring free energy landscapes of chemical reactions* (2006) *Acc Chem Res*, 39:73-81 (citation=248)

(2) Laio, A. et al. *Metadynamics: a method to simulate rare events and reconstruct the free energy in biophysics, chemistry and material science* (2008) *Rep Prog Phys*, 71:291-312 (citation=755)

Since we do observe that loop B and C are extremely flexible in both crystal structure and unbiased MD simulations. It is convincing and reasonable to use them as CVs.

Also the subsequent observations are flawed by the same issue. The correct usage of metadynamics would be to wait for convergence, which on a 2D space could be not straightforward, and to derive considerations on the reconstructed free energy surface, rather than commenting on individual events observed in a biased trajectory and ascribing them to a possible induced fit effect.

Response: As explained above, our CVs have been well selected. We are always careful of our simulations. Our metaMD converged very well as indicated by the following plot. However, it is probably not suitable to show this basic information in this experimental dominated story.

We agree with the reviewer that the FES is one of the most interesting parameters for metaMD. We have investigated the FES for two different free energy states in our previous work: Yuan, S. *et al.* Activation of G-protein-coupled receptors correlates with the formation of a continuous internal water pathway (2014) *Nat Commun*, 5:4733 (citation=57)

However, in this work, we are only interested in how the structure changes upon ligand binding and product leaving, instead of the free energy surface of different states. Thus, we only focus on the process of ligand binding/unbinding as shown in movies. We only show this plot to the review as following:

In the main text, they say that "In apoCCHs, E157 is protonated and E185 is deprotonated, which has been confirmed by pKa calculations on the MD trajectory", but in the Methods they say that "All amino acid residues of the protein were modelled according to their protonation state at neutral pH", so how do they estimate pKa of titratable residues in the MD runs? How therefore are derived the data in Supp. figure 22?

Response: The protonation states of each residue is fixed during MD simulation. We first extracted a series of structures from MD simulation, then submitted each structure to PROPKA for pKa calculation. Based on the calculated results, we made statistics in Supplementary Figure 22. We have clarified the misunderstanding in the Method section.

The protein was put in a water box 16A away from the protein, but this distance should be monitored during the dynamics, unless they put some restraints on translation or rotation of the system.

Response: We don't add any restrain on the system. 16 A is an initial distance. During MD simulations, the box size will fluctuate to some extent since every atoms are moving all the time during simulations. However, the box shape is kept by the periodic boundary conditions (PBC) within the MD simulation tools. This is a standard procedure for modern

MD simulation.

"(NVT ensemble, 1bar)" is evidently a mistake.

***Response:* We corrected this accordingly in the Method part.**

"the simulation temperature was 310 K, and the sampling temperature ΔT was 298 K" please clarify.

***Response:* In the equation 1 of the Method section, T is the temperature of the system. The CVs sample an ensemble at a temperature $T+\Delta T$ which is higher than the system temperature T (well-tempered metaMD). The parameter ΔT can be chosen to regulate the extent of free-energy exploration: $\Delta T=0$ corresponds to standard unbiased molecular dynamics, $\Delta T\rightarrow\infty$ corresponds to standard metadynamics. We add this information to the Method section as well.**

"The convergence of our simulations was judged by using the free energy difference between states X and A at 10 ns intervals. Once the resulting data remained stable over time, the simulation was considered as converged. Each metadynamics simulation lasted 120 ns, and the results were analyzed upon convergence." Many things in this sentence should be rephrased and clarified, also because convergence is probably the most delicate aspect of metadynamics, as in many other enhanced sampling approaches. How precisely are defined X and A? how many recrossings have been observed? which is the threshold used for convergence? etc..

***Response:* We rephrased this sentence accordingly in the SI file. The converge of well-tempered metaMD indicated the Gaussian height. As indicated by above, our metaMD converged very well.**

State X is a reference state, while A is the state after additional biases added to state X. If the Gaussian height is close to zero between state A and X, the simulation is converged as shown by the plot mentioned above. We also added this information to the Method section.

Overall, I should recommend a much clearer presentation of the logical flow of the work and a more careful and appropriate usage of the computational tools.

***Response:* We thank the reviewer's comments again, and we clarified all issues accordingly.**

Reviewer #3 (Remarks to the Author):

GyrI-like proteins have been reported to confer antibiotic resistance through binding of small molecules. In this report, Yuan and co-workers report the catalytic function of GyrI-like proteins that hydrolyzes the cyclopropyl ring of cytotoxic cyclopropanoids. The authors initially identified this catalytic function through gene deletion study, which revealed the disappearance of two hydrolysis product peaks in the ytkR7 gene knock out mutants. Subsequent characterizations of YtkR7 homologues showed that many of the GyrI-like proteins have the same capability to hydrolyze cyclopropanoids. In addition, the authors conducted structural characterization of lin2189, which is a YtkR7 homolog that can hydrolyze multiple cyclopropanoid substrates. While the apo structure of this protein has been previously reported, the authors succeeded in acquiring the co-crystal of 1 with catalytically inactive mutant lin2189. Protein-ligand docking was used to predict the binding of 1 and 2 to lin2189 in the catalytically relevant form. In addition, metaMD simulations were performed to observe the potential movement in loops B and C for substrate entrance and exit. Finally, the authors confirmed that the cytotoxicity of the hydrolyzed product is much lower than the cyclopropanoids and the growth of *E. coli* in presence of 1 is improved upon expression of the cyclopropanoidcyclopropyl hydrolases. Finally, SPR analysis was performed to show that three GyrI-like proteins, SbmC, Rob and YtkR7, are capable of binding antibiotics as one of the detoxification methods. Overall, the work presented is clear and well rounded. Although the structure of the protein has been reported previously, additional computational modeling revealed additional information regarding the catalytic properties of the GyrI-like proteins. Based on the importance of the GyrI-like protein in antibiotic resistance, the reviewer judges that this work is of interest to broad audience and is suitable for publication in Nature Communications with minor revision.

1. Because this is the first example for the catalysis of GyrI proteins, it is informative if the authors provide the kinetic values (K_m and k_{cat}) of one of the identified cyclopropyl ring hydrolases. The enzymatic traits such as pH and temperature dependency should be also presented.

Response: Because the substrates (yatakemycin and CC-1065) are difficult to dissolve in the reaction buffer, we need to add DMSO (25% V/V, final) into the reaction system. Despite many efforts, we failed to obtain satisfactory values for V_{max} and K_M . For example, we obtained the following fitting curves.

2. Can the authors explain how they find the key acidic residues E157 and E185 before crystallization briefly?

Response: In 2004, Anantharaman *et al.* defined the GyrI-like family proteins. Based on the sequence of the MerR-like transcription activator BmrR, they found that the residue E185 (for lin2189) is highly conserved. [Anantharaman, V. and Aravind, L. (2004) *Proteins*, 56: 795-807.]

lin2189 E185 corresponds to BmrR E253. The BmrR E253 influences the ligand binding spectrum. Moreover, *in vitro* transcription data show that E253A and E253Q are constitutively active. [Newberry, K. J., *et al.* (2008) *J Biol Chem*, 283: 26795-26804.]

We thus mutated lin2189 E185, and found that it lost the enzymatic activity. Subsequently, we screened lin2189 E185L-YTM complex, and found that the electron density map of YTM is not clear enough. But from the information gained from the complex structure, we designed the E157 and E185 double mutant.

To make it clear, we added “Previous structural and biochemical studies of BmrR revealed that the glutamate residue Glu-253 has a noticeable effect on its ligand binding and transcription activation¹⁷. This acidic residue is highly conserved among the GyrI-like proteins¹¹, we thus constructed a lin2189 variant (E185L). Interestingly, this variant lost the enzymatic activity and allowed us to obtain a complex structure of lin2189 E185L and 1 in which the electron density for the substrate is partially visible. Detailed investigation of the electron density map implicated that several residues in the solvent accessible channel (e.g., E157) potentially interfered with substrate binding. Therefore, we designed a series of variants based on E185L. As expected, we successfully solved the complex structure of a catalytically-deficient form lin2189 E157A E185L and 1 with complete density at a resolution of 1.2 Å.” to the main text.

3. The authors report that 1694 GyrI proteins that has similar catalytic residues as lin2189. Are the genes encoding them located close to yatakemycin gene cluster? Can the authors predict the substrate for these enzymes from the point of view of the bioinformatic aspect?

Response: From our enzymatic assays with the selected CCHs, we only found YtkR7, C10R6, and SHJG_8481 are, respectively, close to yatakemycin (YTM), CC-1065, and a cryptic biosynthetic gene cluster, which is homologous to that of YTM. For other tested CCHs, we did not observe any such biosynthetic gene clusters. These 1696 proteins are much more like evolutionarily conserved proteins rather than those acquired by horizontal gene transfer, and thus they should play beneficial roles to their hosts. Here, we show that they have the ability recognize YTM and CC-1065. It might be an accidental event during protein

evolution. We regard these GyrI-like proteins as an intracellular scavenger for xenobiotics.

4. How do the cyclopropyl ring hydrolases bind with the antibiotics except yatakemycin? Do they also bind with the substrate-binding cage? The authors should explain the difference of the binding mode between the cyclopropyl ring hydrolases and the reported GyrI protein structures.

Response: The GyrI-like proteins possess a duplicate $\beta\alpha\beta\beta$ configuration and appear to have been adapted for small-molecule binding. They each possess a different solvent accessible cage. Structural studies of BmrR for drug binding and multispecificity reveals that aromatic side chains lining the drug pocket form a rigid, aromatic platform for diverse drugs to dock. Strange but comprehensible is that our tested GyrI-like have the ability to bind structurally unrelated compounds. The different set of aromatic and hydrophobic residues might determine their substrate spectrum. We have added some discussion in the text.

5. Last sentence on page 5 seems to imply that substrates other than 1 and 2 were used for the in vitro and in vivo studies, but it is unclear which molecules were actually tested.

Response: We only tested the two substrates (yatakemycin and CC-1065). From our assays, we found that all the tested CCHs can hydrolyze yatakemycin, which is also consistent with its higher DNA alkylating efficiency. However, only five of the tested CCHs hydrolyze CC-1065. Recently, we have engineered a strain in a modified CC-1065 producing strain and isolated gilvusmycin (related to CC-1065) and its cyclopropyl hydrolyzed product, which indicates that the protein C10R6 recognize gilvusmycin. Moreover, from figure 1a, we can see that duocarmycins are similar to the left part of CC-1065, so it is predictable that duocarmycins are proper substrates of CCHs. So we will see at least this family of compounds can be as the substrates.

6. Did the authors observe the formation of the spontaneously oxidized product of 7?

Response: From the HPLC profile of fermentation products, we did not observe the formation of the spontaneously oxidized product of 7, which indicates 7 is more stable than 5.

7. Isolated compounds do not appear to be entirely pure based on the NMR spectra provided.

Response: We finally tried to purify all the compounds with semi-preparative HPLC. Combined with HR-MS/MS, we could observe the necessary signals from the NMR spectra, so we can elucidate the structures of compounds 5, 6, and 7.

8. The HPLC peak intensities of the assays comparing the activities of the mutants are different for the double mutant compared to the other single mutants. Is there any reason for this difference?

Response: Sorry for the misunderstanding. Because these are two batches of enzymatic assays, we added about 2-3 times more substrate when using lin2189 E157A E185L.

Reviewers' Comments:

Reviewer #1 (Remarks to the Author):

The revised manuscript addresses major concerns raised by the reviewer. The discovery and characterization of the new class of GyrI-like proteins are novel, important and have implications regarding our understanding of bacterial resistance mechanisms and discovery of new antibiotics (as well as antibiotic targets). As such, the reviewer recommends that the manuscript is accepted.

The reviewer has only a few comments...

The authors make the statement "Seven of these GyrI-like proteins were chosen from a variety of bacteria and archaea (from diverse environments, including soil and the human microbiota: SHJG_8481, Chte2144, ETI84332.1, C10R6, lin2189, SSDG_03674, and MA1133) and tested for their enzymatic activity, and all were found to convert 1 to 5 (Fig. 1e, Supplementary Figs. 10 and 11), suggesting that a vast number of prokaryotic GyrI-like proteins can be classified as CCHs."

The authors use the word "vast" a number of times in the manuscript. This may be a bit of an overstatement. In the context of this sentence, it certainly is. It is likely that many other members have different functions.

The following statement is a bit confusing, "The finding that these distantly-related GyrI-like proteins each have the ability to bind xenobiotics suggests that organisms use these proteins for detoxification for a very long time."

I believe that the authors are stating that using the GyrI-like proteins for detoxification appears to be an ancient strategy?

Reviewer #2 (Remarks to the Author):

The Authors partially answer the issues raised in the review.

From the rebuttal it emerges a possible misunderstanding concerning the issue of the usage of PCA on a variable which has been accelerated via metaMD. If the Authors did the PCA on a plain MD and then, based on the results, they decided which variables should be accelerated, then this would be fine. If this is the case, I would recommend explicitly saying this in the

manuscript : " Principal component analysis made on unbiased MD simulation showed that both loop B and loop C undergo noticeable fluctuations...." If, in contrast, the Authors have performed the PCA over the biased trajectories resulting from the application of the MetaMD, then my objections persist.

One more comment, in the well tempered metaMD the height of the hills tends to decrease along time by construction and irrespective of the fact that actual convergence is reached. So other criteria for convergence should be considered, like the number of recrossing events observed and the diffusive behavior of the dynamics.

Finally, while it is true that in "modern" MD the simulation box shape is preserved and PBC are used, there is still the possibility, due to a possible translational or a rotational drift, that the simulated solute gets in vicinity of the box border itself, giving rise to unwanted spurious effects. A check on the minimal distance between the solute and the box border is advisable.

Reviewer #3 (Remarks to the Author):

The authors report the catalytic function of GyrI-like proteins that hydrolyzes the cyclopropyl ring of cytotoxic cyclopropanoids. YtkR7 and its homologues are thought to be responsible for binding and hydrolyzing DNA-alkylating agents such as yatakemycin to confer resistance in the producing organism. Together with the finding that GyrI-like proteins are widely distributed among prokaryotes and eukaryotes suggest that these proteins likely play an important role, possibly in detoxification of xenobiotics. While the reviewer is not familiar with the details of the computational methods utilized in this study, the biochemical aspects of the work are reasonable. Although the authors did not perform additional experiments requested by the reviewer such as the kinetics analysis, an acceptable explanation was provided. One issue that the reviewer has with the authors' response is that not all the enzymatic assays with the wild type and the mutant enzymes were performed using the same assay conditions. For instance, it is unclear why the concentration of the substrate was altered in the assay performed with the double mutant. This explanation is also missing from the method section explaining the enzymatic assays performed, which should be corrected to incorporate the differences in assay conditions. Overall the reviewer judges that the conclusion drawn based on the biochemical work presented is acceptable, and this article is acceptable to be published in Nature Communications, as long as all the issues with the computational work have been properly addressed.

Responses to Reviewers' Comments:

Reviewer #1 (Remarks to the Author):

The revised manuscript addresses major concerns raised by the reviewer. The discovery and characterization of the new class of GyrI-like proteins are novel, important and have implications regarding our understanding of bacterial resistance mechanisms and discovery of new antibiotics (as well as antibiotic targets). As such, the reviewer recommends that the manuscript is accepted. Response: We thank the reviewer for the recommendation.

The reviewer has only a few comments...

The authors make the statement "Seven of these GyrI-like proteins were chosen from a variety of bacteria and archaea (from diverse environments, including soil and the human microbiota: SHJG_8481, Chte2144, ETI84332.1, C10R6, lin2189, SSDG_03674, and MA1133) and tested for their enzymatic activity, and all were found to convert 1 to 5 (Fig. 1e, Supplementary Figs. 10 and 11), suggesting that a vast number of prokaryotic GyrI-like proteins can be classified as CCHs." The authors use the word "vast" a number of times in the manuscript. This may be a bit of an overstatement. In the context of this sentence, it certainly is. It is likely that many other members have different functions.

Response: We agree with the reviewer. There were 12, 304 GyrI-like sequences in databases (as of September 2016), and in this case we can use "vast" to describe the huge number. Our study indicates that a subfamily of these proteins (1, 696 sequences) can be classified as cyclopropanoid cyclopropyl hydrolases, and here we have changed the word "vast" to "great" to describe the

number, that is, "....., suggesting that a great number of prokaryotic GyrI-like proteins can be classified as CCHs."

The following statement is a bit confusing, "The finding that these distantly-related GyrI-like proteins each have the ability to bind xenobiotics suggests that organisms use these proteins for detoxification for a very long time."

I believe that the authors are stating that using the GyrI-like proteins for detoxification appears to be an ancient strategy?

Response: We agree with the reviewer's suggestion. We have rewritten the sentence "The finding that these distantly-related GyrI-like proteins each have the ability to bind xenobiotics suggests that using the GyrI-like proteins for cellular detoxification appears to be an ancient strategy."

Reviewer #2 (Remarks to the Author):

The Authors partially answer the issues raised in the review.

From the rebuttal it emerges a possible misunderstanding concerning the issue of the usage of PCA on a variable which has been accelerated via metaMD. If the Authors did the PCA on a plain MD and then, based on the results, they decided which variables should be accelerated, then this would be fine. If this is the case, I would recommend explicitly saying this in the manuscript : " Principal component analysis made on unbiased MD simulation showed that both loop B and loop C undergo noticeable fluctuations...." If, in contrast, the Authors have performed the PCA over the biased trajectories resulting from the application of the MetaMD, then my objections persist.

Response: Probably there is a misunderstanding here. The PCA analysis has been performed on the nonbiased MD simulations instead of metaMD. MetaMD was only used to sample the substrate entrance pathway and the product leaving pathway. We further clarified this explicitly in the Method section of MD simulation in the second revised version.

One more comment, in the well tempered metaMD the height of the hills tends to decrease along time by construction and irrespective of the fact that actual convergence is reached. So other criteria for convergence should be considered, like the number of recrossing events observed and the diffusive behavior of the dynamics.

Response: We agree with the reviewer. We are always careful about our simulations and checked all details very carefully.

Finally, while it is true that in "modern" MD the simulation box shape is preserved and PBC are used, there is still the possibility, due to a possible translational or a rotational drift, that the simulated solute gets in vicinity of the box border itself, giving rise to unwanted spurious effects. A check on the minimal distance between the solute and the box border is advisable.

Response: We thank the reviewer's comments. When we analyzed the data, we used the option "pbc wrap" so that each atom in the neighboring box won't mix with the current one. Protein backbone, which had been centered in the PBC box, was superimposed with each other prior to all data analysis.

Reviewer #3 (Remarks to the Author):

The authors report the catalytic function of GyrI-like proteins that hydrolyzes the cyclopropyl ring of cytotoxic cyclopropanoids. YtkR7 and its homologues are thought to be responsible for binding and hydrolyzing DNA-alkylating agents such as yatakemycin to confer resistance in the producing organism. Together with the finding that GyrI-like proteins are widely distributed among prokaryotes and eukaryotes suggest that these proteins likely play an important role, possibly in detoxification of xenobiotics. While the reviewer is not familiar with the details of the computational methods utilized in this study, the biochemical aspects of the work are reasonable. Although the authors did not perform additional experiments requested by the reviewer such as the kinetics analysis, an acceptable explanation was provided. One issue that the reviewer has with the authors' response is that not all the enzymatic assays with the wild type and the mutant enzymes

were performed using the same assay conditions. For instance, it is unclear why the concentration of the substrate was altered in the assay performed with the double mutant. This explanation is also missing from the method section explaining the enzymatic assays performed, which should be corrected to incorporate the differences in assay conditions.

Response: We agree with the reviewer's suggestion. We have revised the part in the Method section (the concentration of the substrate used in the assays: 100-300 \$\mu\$ M YTM (or CC-1065)).

Overall the reviewer judges that the conclusion drawn based on the biochemical work presented is acceptable, and this article is acceptable to be published in Nature Communications, as long as all the issues with the computational work have been properly addressed.

Response: We thank the reviewer for the recommendation.